# Modulation of Serum Metabolic Profiles by *Bifidobacterium breve* BBr60 in Obesity: A Randomized Controlled Trial

**DOI:** 10.3390/foods13223655

**Published:** 2024-11-17

**Authors:** Ying Wu, Dejiao Gao, Yujia Pan, Yao Dong, Zhouya Bai, Shaobin Gu

**Affiliations:** 1College of Food and Bioengineering, Henan University of Science and Technology, Luoyang 471000, China; wuying2000@126.com (Y.W.); gaodej522@163.com (D.G.); 17739736258@163.com (Y.P.); 2Henan Engineering Research Center of Food Microbiology, Luoyang 471000, China; 3Germline Stem Cells and Microenvironment Lab, College of Animal Science and Technology, Nanjing Agricultural University, Nanjing 210095, China; yao.dong@stu.njau.edu.cn; 4Henan Engineering Research Center of Food Material, Henan University of Science and Technology, Luoyang 471023, China

**Keywords:** *Bifidobacterium breve* BBr60, overweight or obesity, serum metabolism, metabolic pathway, randomized controlled trial

## Abstract

Obesity, a prevalent metabolic disorder in youth, leads to complications and economic strain. Gut dysbiosis significantly contributes to obesity and metabolic issues. *Bifidobacterium breve*, a probiotic strain, may help regulate gut dysbiosis and benefit obese individuals. However, more research is needed on its effect on serum metabolism. A total of 75 overweight or obese young adults (aged 19–45) participated in the current study, and were randomly divided into probiotic and placebo groups using a random number table. Both groups received dietary guidance and underwent twelve weeks of intervention with either oral *Bifidobacterium breve* BBr60 (BBr60) or a placebo. After the intervention, collection and analysis of fasting serum samples were conducted using mass spectrometry coupled with liquid chromatography. Analyses of associations were conducted in order to determine the correlations between key serum metabolites and clinical obesity indicators, aiming to understand the influence of BBr60. Due to 10 participants dropping out for personal reasons, the study included 32 and 33 participants in the placebo and the BBr60 groups, respectively. The BBr60 intervention significantly regulated 134 serum metabolites and influenced crucial metabolic pathways in obesity management (*p* < 0.05), including ascorbate and aldarate metabolism for oxidative stress reduction, cholesterol metabolism for lipid regulation, parathyroid hormone synthesis, secretion and action for the endocrine system, oxidative phosphorylation for enhanced energy efficiency, and glycolysis/gluconeogenesis for glucose metabolism. Analysis showed a positive relationship between fasting blood glucose (FBG), aspartate aminotransferase (AST), total protein (TP), and the content of 5-Methyl DL-glutamate (*p* < 0.05). Similarly, body mass index (BMI), weight, and body fat percentage (BFP) were positively linked to serum metabolites (1-Hydroxycyclohexyl) acetic acid, and 5-Oxooctanoic acid (*p* < 0.05). Significant associations of AST levels with key serum metabolites in cholesterol metabolism pathways further suggest BBr60’s potential to improve liver function and overall metabolic health in overweight or obese individuals. These findings support BBr60’s effectiveness in modulating serum metabolic profiles and suggest it may improve liver function and BMI in overweight or obese individuals by regulating key serum metabolites.

## 1. Introduction

As economic, social, and lifestyle changes occur, increasing treads of overweight and obesity is a common condition of populations living in environments with abundant calories and low physical activity [1]. At present, the number of overweight or obese individuals among people older than 5 years is 2.6 billion in 2020, accounting for 38% of the global population, and this number of people will climb to more than 4 billion in 2035, representing a majority of the world’s population [2]. The adverse effects of overweight or obesity affect every function of the body, contributing to many comorbidities, including hyperlipidemia, type 2 diabetes, hypertension, and others [3]. Growing evidence has demonstrated that the occurrence of overweight/obesity is tightly linked to the homeostatic regulation of comprehensive metabolism [4,5]. The circulation of blood metabolites potentially impacts and exhibits the host metabolism and progression of obesity [6]. Based on existing evidence, the metabolic properties associated with obesity have been identified to include amino acids, glycerol, and glycerophosphocholines [7,8]. In a previous study, 49 obesity metabolites, including 23 lipids, 14 amino acids, 3 nucleotides, 3 peptides, and others, have been found and showed significant association with the vital indicator of obesity (body mass index, BMI) [7].

In recent studies, it has been found that overweight/obesity-associated circulating metabolites are also closely linked to gut microbes; in addition, intestinal bacteria have a significant effect on an overweight/obese adult’s metabolic health [9]. There have been many studies demonstrating that the bacterial composition of the intestines differs significantly between individuals with a normal weight and those who are overweight or obese [10]. Microbiome changes linked to overweight/obesity are commonly known as “dysbiosis”, which refers to any unfavorable change in gut microbiota structure and function, and is caused by a lack of beneficial function or the presence of harmful microbial activity [11,12]. The microbiome can be modulated with a complementary intervention resource, and probiotics are defined as “live microorganisms” and could effectively modify misbalanced microbiota to ameliorate key obesity-related parameters with certain doses of probiotic strains [13,14]. The primary probiotic genera include *Bifidobacterium*, *Lactobacillus*, *Enterococcus*, *Saccharomyces*, and *Saccharomyces boulardii*, according to current studies [15,16]. Among these genera, *Bifidobacterium* species are frequently utilized in functional foods and dietary supplements due to their proven benefits in managing metabolic disorders. Specifically, *Bifidobacterium* was chosen for our study because of its documented efficacy in modulating gut microbiota and improving metabolic profiles relevant to obesity [17,18]. The strain of *Bifidobacterium breve* BBr60 (BBr60) has been shown to enhance gut barrier integrity, modulate lipid metabolism, and reduce inflammatory responses in obese conditions [19], making it an ideal candidate for investigating potential benefits for overweight or obese individuals. Administering the *Bifidobacterium animalis* IPLA R1 strain resulted in significant reductions in levels of stearic and eicosatetraenoic fatty acids compared to sole administration of a high-fat diet (HFD). Additionally, it significantly upregulated the genes Acox1, Ppargc1a, Cpt1a, and Hahd, thereby promoting alterations in lipid metabolism and glucose homeostasis [20]. Several studies have demonstrated that *Bifidobacterium* species are capable of producing exocellular carbohydrate polymers (EPSs) which can serve as fermentable substrates, thereby enhancing the proliferation of gut microorganisms to produce unconjugated bile acids. This deconjugation process may lead to a reduction in bile acid reabsorption. As a result, the liver synthesizes new bile acids from cholesterol, which results in a decrease in cholesterol levels in the body [21]. Given BBr60’s impact on the microbiota—specifically, its association with significant improvements in vital intestinal bacteria such as *Klebsiella*, *Bacteroides*, and *Dialister*—it has demonstrated greater efficacy than the placebo in reducing weight and BMI in preliminary studies [19]. This compelling evidence supports the need to further explore the eradication efficacy and safety of *Bifidobacterium breve* BBr60 over a placebo in treating metabolic disturbances associated with overweight and obesity.

Therefore, the trials which assessed the comparative effectiveness of *Bifidobacterium breve* BBr60 on regulating the disturbed metabolic profiling in overweight/obese young participants in China compared to the placebo were randomized, double-blind, and placebo-controlled. A total of 75 overweight/obese adult patients were equally likely to receive *Bifidobacterium breve* BBr60 or a placebo for 12 weeks, and briefly educated regardingdietary recommendations and total daily energy intake (1800 kcal). Serum samples were collected for a metabolomics assay before and after treatment to access the effectiveness of regulating metabolic disturbance of overweight/obesity. Ultimately, there was a scientific basis for BBr60’s clinical application in short-term diets and obesity prevention/intervention (presented in Graphical Abstract).

## 2. Materials and Methods

### 2.1. Design and Population of Study

An experimental design was used in this study; it was a randomized, double-blind, placebo-controlled study, which was conducted in the School of Food and Bioengineering at Henan University of Science and Technology during the period between June 2023 and March 2024. The protocol was carried out according to the Declaration of Helsinki by the World Medical Association. It was approved by the Ethics Commission of the First Affiliated Hospital of Henan University of Science and Technology (NCT06305650, 13 January 2024. https://www.chictr.org.cn). Then, all participants signed informed consent forms under the following inclusion criteria: (1) aged 19–45 years; (2) BMI ≥ 28 kg/m^2^; and (3) participants understand the research procedure and agree to participate after signing the written consent form. The exclusion criteria included (1) intake of probiotics during the last 3 months; (2) administration of antibiotics, laxatives, or other dietary supplements in the past 30 days; (3) subjects with actual or suspected alcohol or drug abuse; (4) severe medical conditions (e.g., renal or hepatic disease, neurological disorders); (5) pregnant, lactating, or likely to become pregnant and unwilling to use contraception. To be eligible, participants had to meet all of the criteria stated above.

A random assignment was made for placebo (*n* = 36) or probiotic group (*n* = 39) by a random number table. Participants of placebo group received daily maltodextrin (3 g), and participants of probiotic group ingested daily probiotic powder (1 × 10^10^ CFU daily, Wecare Probiotics Co., Ltd., Suzhou, China) for 12 weeks. It is advisable to consume probiotics or placebo 30 min after meals, accompanied by warm water or milk, with the water temperature not exceeding 40 degrees Celsius. A daily intake registration card should be maintained, where the date should be noted and a tick mark provided. Additionally, it is important to retain the finished packaging for recycling purposes. While taking probiotics, if there is a need to consume antibiotics, an interval of 2–3 h should be observed between the two. In the end, 65 participants completed all of the experiments (Figure 1). It is recommended that all participants reduce their daily energy intake to 1800 calories and participate in nutrition education programs, including the occurrence risks and pathogenesis of overweight/obesity, dietary recommendations, principles of weight loss and restrictions, and energy intake and associated costs.

### 2.2. Serum Metabolomic Analysis

Before and after treatment, morning blood samples were collected after at least 12 h of overnight fasting at each outpatient visit and centrifuged (15 min, 3000× *g*) to separate serum samples in metabolomics analysis in weeks 0 and 12. A 100 μL serum sample was mixed with quadruple extraction solution, and extraction solution contained MeOH, CAN at ratio of 1:1, and deuterated internal standards. After vortexing for 30 s and sonicating for 10 min at 4 °C, the mixture was cooled down and precipitated proteins were incubated for 1 h at −40 °C. Analysis of the supernatant was performed in a new glass vial after centrifuging at 13,800× *g* for 15 min at 4 °C. In order to obtain a quality control (QC) sample, equal parts of the supernatants were combined.

LC-MS/MS was conducted utilizing a UHPLC system ((Vanquish, Thermo Fisher Scientific, San Jose, CA, USA) equipped with a Waters ACQUITY UPLC BEH Amide column (2.1 mm × 50 mm, 1.7 μm) and interfaced with an Orbitrap Exploris 120 mass spectrometer (Orbitrap MS, Thermo Fisher Scientific, San Jose, CA, USA). In the mobile phase, 25 mmol/L ammonium acetate and 25 mmol/L ammonium hydroxide were dissolved in water (pH = 9.75) and acetonitrile was used as the solvent. The auto-sampler was kept at 4 °C with an injection volume of 2 μL. By using Xcalibur software (version 4.1, Thermo Fisher Scientific, San Jose, CA, USA), Orbitrap Exploris 120 mass spectrometers were operated in IDA mode, continuously assessing the MS spectrum over time. The ESI source was configured with a sheath gas flow of 50 Arb, auxiliary gas flow of 15 Arb, and a capillary temperature of 320 °C. The full MS and MS/MS resolutions were set at 60,000 and 15,000, respectively, with collision energy at SNCE values of 20, 30, and 40. The spray voltage was set at 3.8 kV and −3.4 kV for positive and negative ion mode, respectively.

In order to analyze the data, ProteoWizard 3.0 was used to convert the raw data to the mzXML format and an in-house program developed in R was used to analyze the data subsequently. R and BiotreeDB (version 3.0) were used for metabolite identification.

### 2.3. Statistical Analysis

The mean and standard deviation (SD) of quantitative data were expressed as follows: for normally distributed data, a two-tailed *t*-test was used to evaluate statistical significance. For datasets that deviate from a normal distribution, the Mann–Whitney U test and the Wilcoxon signed-rank test were employed for inter-group and intra-group comparisons, respectively. The Chi-squared test was used to evaluate statistical significance of categorical variables represented as frequencies (percentages). Orthogonal partial least squares discriminant analysis (OPLS-DA) and principal component analysis (PCA) were performed with SIMCA software (version 18.0.1, Sartorius Stedim Data Analytics AB, Umea, Sweden). KEGG (http://www.genome.jp/kegg/) and MetaboAnalyst (http://www.metaboanalyst.ca/) were used for pathway analysis. The KEGG enrichment analysis of differential metabolites was performed using Fisher’s exact test. A threshold of *p* < 0.05 was established to make sure that the observed differences were statistically significant, thereby enhancing the interpretative value of the results.

## 3. Results

### 3.1. Characteristics of the Study Population

Appendix A details the participant characteristics, including mean age, sex distribution, baseline BMI, and others. The mean age was 27.88 years and 30.38 years for the BBr60 and placebo groups, respectively. The BBr60 group had 72.7% females, while the placebo group had 59.4%. The values of baseline BMI were 30.80 kg/m^2^ and 31.96 kg/m^2^ for the BBr60 and placebo groups, respectively. The values of Waist-to-Hip Ratio (WHR) were 0.99% and 1.01% in the BBr60 and placebo groups, respectively. And mean age, sex distribution, baseline BMI, and WHR all did not demonstrate a statistically significant difference between the groups. Additionally, no significant differences were analyzed between the two groups in other indictors of renal and hepatic function, body composition, and biochemical parameters.

### 3.2. Effect of BBr60 on Metabolic Profiles After the 12-Week Intervention

The analysis of clinical indicators showed that BBr60 significantly influenced BMI, body composition, blood lipid levels, liver function, kidney function, and fasting blood glucose levels after a 12-week intervention compared to the placebo group, as detailed in Appendix A. In addition, the serum metabolic effects of BBr60 were also examined in individuals with overweight/obesity. Overall distribution of BBr60-before, BBr60-after, placebo-before and placebo-after have been presented in the three-dimensional PCA (3D-PCA) provided with Appendix A. The comparison between BBr60-before and BBr60-after have been analyzed in our previous study [19]; the current study was focused on analyzing the regulating effect of BBr60 compared to the placebo in the twelfth week. OPLS-DA was utilized to identify relevant biomarkers (variable influence on projection, VIP > 1) between the BBr60 and placebo groups in the twelfth week. A separation was observed in the score plot of OPLS-DA (R2Y = 0.50 and Q2 = −0.45) between samples from the BBr60 and placebo groups (Figure 2A,B). The serum metabolic profile was presented by three-dimensional PCA (3D-PCA) with a cumulative proportion of principal component 1, 2, and 3 accounting for 72.6% of the total components. Statistical analysis revealed clear distinctions in the metabolic profiles between the BBr60 and placebo groups, with significant differences in metabolite regulation observable in Figure 2C,D. A total of 93 serum metabolites were significantly up-regulated and 41 serum metabolites were significantly down-regulated in the BBr60 group compared to the placebo group, as shown in Figure 2D. Detailed information about these critical serum metabolites is provided in Appendix A.

Furthermore, the top 10 differentiating substances with significant up-regulation and down-regulation were further analyzed between the BBr60 and placebo groups. They are categorized under 13 classes, including alcohols and polyols, amines, amino acids, peptides, analogues, aminoxides, diphenylmethanes, fatty acids and fatty acid esters, conjugates, fatty amides, glycerophosphocholines, imidazolidines, purines and purine derivatives, quaternary ammonium salts, pyrimidines and pyrimidine derivatives, amines, glycerophosphocholines, and quaternary ammonium salts (Figure 3A). Among the differentiating substances, all serum metabolites categorized as amines showed a statistically significant increase in response to BBr60 treatment, including Tetradecylamine, 1-Hexadecylamine, (3-Methylbutyl) (6-methylheptan-2-yl) amine, and 1-Octadecylamine (Figure 3B). The glycerophosphocholines, such as 1-O-Hexadecyl-sn-glycero-3-phosphocholine and 1-O-Octadecyl-sn-glyceryl-3-phosphorylcholine, were significantly up-regulated by BBr60 intervention, with the exception of 2-(5-Oxovaleryl) phosphatidylcholine. In contrast, other lipid metabolites including Hexenoylcarnitine (Car(6:1)), 5-Oxooctanoic acid, and 4-Oxo-4-((3-oxodecan-2-yl)amino) butanoic acid demonstrated significant down-regulation due to BBr60 treatment. Additionally, there was a significant increase in the relative abundance of quaternary ammonium salts such as dihexadecyldimethylammonium cations, tetrapropylammonium cations, and tetraoctylammonium cations in the BBr60 group. In all, BBr60 intervention presented an obvious effect on serum metabolism; 134 serum metabolites were significantly regulated by BBr60 compared to placebo intervention. A total of 20 serum metabolites were identified as vital markers and mainly categorized under amines, glycerophosphocholines, and quaternary ammonium salts.

### 3.3. The Effect of BBr60 on Metabolic Pathways After the 12-Week Intervention

Notable changes in serum metabolites (VIP > 1, *p* < 0.05) were identified between the placebo and BBr60 groups, enabling the identification of key pathways involved in 20 metabolic processes (Appendix A). Upon BBr60 treatment, 15 metabolic pathways (*p* < 0.1) exhibited changes compared to the placebo intervention (Figure 4A). Of these, six metabolic pathways showed statistical significance (*p* < 0.05) and were influenced by BBr60. Notably, cholesterol metabolism was markedly up-regulated, demonstrating significant modulation in 6.67% of differential metabolites, highlighting its crucial role in lipid homeostasis and implications for cardiovascular health in obesity. Ascorbate and aldarate metabolism, crucial for managing oxidative stress and enhancing metabolic health, showed the lowest *p*-values among the significantly altered pathways. Additionally, oxidative phosphorylation, important for energy metabolism, also showed significant modulation, suggesting potential benefits in increasing metabolic rate and energy expenditure in obese individuals. Despite equal up- and down-regulation, mineral absorption involved 13.33% differential metabolites, showing significant regulation but no discernible trend in changes. Furthermore, pathways such as parathyroid hormone synthesis, secretion, and action, and choline metabolism in cancer were also significantly altered, reflecting broader metabolic disturbances related to obesity.

The metabolic pathways of parathyroid hormone synthesis, secretion, and action, as well as oxidative phosphorylation, were significantly modulated by BBr60, with a 6.67% differential in metabolites compared to all identified metabolites within these pathways. These pathways are integral to the endocrine system and energy metabolism. Further analysis revealed that BBr60 significantly regulated six metabolic pathways when compared to the placebo intervention. Notably, the pathways involved in ascorbate and aldarate metabolism, cholesterol metabolism, parathyroid hormone synthesis, secretion and action, and oxidative phosphorylation were significantly affected and warrant further investigation due to their critical roles in metabolic health.

### 3.4. The Association of Vital Serum Metabolites and Clinic Indicators in Overweight or Obese Individuals

To analyze the potential relationship between vital serum metabolites and clinic indicators, correlation analysis was conducted (Figure 5A,B). The top 10 differentiating substances with significant up-regulation and down-regulation were used to analyze the association between the vital marker and apparent indexes of obesity. It can be discovered that indicators of weight, BMI, and body fat percentage (BFP) have a positive correlation with the serum metabolites of (1-Hydroxycyclohexyl) acetic acid (*p* < 0.05) and 5-Oxooctanoic acid (*p* < 0.05). The level of fasting blood glucose (FBG), significantly influenced by BBr60 treatment, demonstrated a statistically significant positive correlation with the levels of 5-Methyl DL-glutamate (*p* < 0.05). Also, liver function indexes of aspartate aminotransferase (AST) and total protein (TP) exhibited a positive correlation with the serum metabolites of 5-Methyl DL-glutamate (*p* < 0.05), 6-(1-Pyrrolidinyl)-1H-purine (*p* < 0.05). Furthermore, an analysis of correlations was conducted to assess the relationship between serum metabolites involved in the altered metabolic pathways and clinical indicators in individuals with overweight/obesity (Figure 5B). The AST level showed an obvious association with the content of PC (14:0/P-18:1(9Z)) associated with the pathways of linoleic acid metabolism, retrograde endocannabinoid signaling, and choline metabolism in cancer (*p* < 0.05). Also, globular proteins (GLB), as the vital indicator of liver function, showed an obvious association with the content of taurochenodeoxycholic acid involved in the pathways of cholesterol metabolism. These results indicated that those serum metabolites may be key makers that affect clinical factors/disease phenotypes via the pathways of linoleic acid metabolism and cholesterol metabolism.

## 4. Discussion

According to this randomized controlled trial, BBr60’s administration in a population of overweight/obese young adults achieved a significant impact in metabolic profiles compared with the placebo group. A total of 134 serum metabolites and 6 metabolic pathways were significantly regulated by BBr60. Of particular note, the pathways of cholesterol metabolism, mineral absorption, ascorbate and aldarate metabolism, choline metabolism in cancer, parathyroid hormone synthesis, secretion and action, and oxidative phosphorylation were identified as vital metabolic pathways (*p* < 0.05). Association analysis highlighted that fasting blood glucose (FBG) and liver function indices, specifically aspartate aminotransferase (AST) and total protein (TP), exhibited significant positive correlations with the levels of 5-Methyl DL-glutamate. The substantial associations of AST and TP with serum metabolites involved in the linoleic acid and cholesterol metabolism pathways underscore potential biochemical interactions that may influence liver health. Cholesterol metabolism, critical for maintaining lipid homeostasis, can impact hepatic function by modulating lipid accumulation and synthesis in the liver. Enhanced cholesterol metabolism through dietary intervention may help reduce hepatic fat content and thus alleviate the burden on liver functions, potentially decreasing the progression of fatty liver disease, a common comorbidity in obesity. Moreover, linoleic acid, as a polyunsaturated fatty acid, is involved in the production of signaling molecules that can regulate inflammatory responses in the liver. Its metabolism could affect liver health by altering inflammatory pathways that contribute to hepatic steatosis and insulin resistance. Therefore, the significant relationship between liver enzymes and metabolites in these pathways might reflect changes in the liver’s metabolic and inflammatory states, which are crucial for managing obesity-related liver complications.

Significant structural alterations of intestinal microbiota between lean and obese participants have been found in substantial studies and are recognized as a therapeutic target of obesity [22,23]. Consequently, the medical community has increasingly focused on the administration of probiotics as a potential preventative approach to overweight and obesity [24]. A research meta-analysis on randomized controlled trials encompassed 957 participants, with an average BMI of 27.6 kg/m^2^, and intervention durations varying between 3 and 12 weeks. The administration of probiotics resulted in a significantly greater decrease in BMI, weight, and fat percentage (−0.27 kg/m^2^, −0.60 kg, −0.60%, respectively), in comparison to the placebo [25]. In the current study, a total of 60 overweight/obese young adults (19–45 years) were enrolled and received oral probiotics (BBr60) or a basis placebo with intervention for 12 weeks. The outcomes of clinical indicators revealed effective regulation of BBr60 on the indicator of BMI, body composition, blood lipid, liver function, and kidney function. Even if the intervention group also showed some effectiveness, this may be because of the recommended daily diet or exercise pattern. However, the probiotic group can still significantly improve fasting blood glucose on this basis. Additionally, the effectiveness of BBr60 on serum metabolism was further investigated, and the serum metabolic profile was obviously regulated. Other studies also demonstrated relationships between serum metabolome and overweight during pregnancy; some corresponding studies of probiotics in other populations have presented a similar effect on serum metabolism [7,26]. A placebo-controlled study was investigated on 66 overweight and non-diabetic subjects. Over the course of 12 weeks, a combination of *Lactobacillus curvatus* HY7601 and *Lactobacillus plantarum* KY1032 supplemented significantly regulated the metabolomics, with an increase in the levels of carnitine metabolites compared with that of the placebo group [26]. A combination of *B. longum*, *B. breve*, and *B. infantis* significantly increased the concentration of heptadecanoic acids, 3-hydroxybutyrate, erythrose, glucose, arabinose, benzoic acid, and taurine, metabolites associated with glucogenic and ketogenic circuits [27]. A dietary intervention and *Bifidobacterium longum* subsp. *Longum* treatment effectively decreased the contents of L-alanine and L-glutamine in feces, which are associated with the central carbon metabolic pathway in cancer [28]. Supplementation with the *Bifidobacterium* IPLAR1 strain also significantly decreased the levels of stearic and eicosatetraenoic fatty acids than the sole administration of the high-fat diet in mice [20]. Also, BBr60 intervention significantly regulated 134 serum metabolites with 93 up-regulations and 41 down-regulations, compared to placebo intervention in the current study. Twenty serum metabolites were identified as vital markers, including six lipid metabolites; four out of six serum lipid metabolites were significantly down-regulated, such as Hexenoylcarnitine (Car(6:1)), 5-Oxooctanoic acid, 4-Oxo-4-((3-oxodecan-2-yl)amino)butanoic acid, and 2-(5-Oxovaleryl) phosphatidylcholine. In addition, the relative abundance of 5-Oxooctanoic acid, known as a medium-chain fatty acid, exhibited a significant relationship with the level of weight, BMI, and BFP. This result may indicate the potential interaction relationship of 5-Oxooctanoic acid and BFP or BMI in overweight/obese individuals after BBr60 intervention. 5-Methyl DL-glutamate and 6-(1-Pyrrolidinyl)-1H-purine, as amino acids, peptides, and analogues or purines and purine derivatives, have shown significant association with liver function indicators (TP, AST). Evidence suggests that obesity is associated with liver dysfunction markers (ALT and AST) [29]. Moreover, concentrations of hepatic enzymes increased (ALT and AST) with gradually increasing BMI in obesity [30]. There is a direct connection between the metabolites of amino acids and reducing body fat of obesity after *Bifidobacterium* intervention. The combination of a dietary intervention and *Bifidobacterium longum* subsp. *Longum* treatment could reduce the concentrations of amino acids (L-alanine and L-glutamine) and regulate fat metabolism [28]. In the current study, alleviating liver function and BMI in obesity after BBr60 intervention may be associated with the metabolites of 5-Oxooctanoic acid, 5-Methyl DL-glutamate, and 6-(1-Pyrrolidinyl)-1H-purine.

Furthermore, the metabolic pathways were analyzed, with significant alteration of serum metabolites in the BBr60 group, and six metabolic pathways were significantly regulated by BBr60 compared to the placebo intervention. The metabolic pathways of significance that were regulated included mineral absorption, ascorbate and aldarate metabolism, cholesterol metabolism, choline metabolism in cancer, parathyroid hormone synthesis, secretion and action, and oxidative phosphorylation. Notably, crucial metabolites associated with cholesterol metabolism exhibited a significant correlation with the liver function indicator, GLB. Previous research suggests a correlation between liver dysfunction and obesity in humans [29]. The homeostasis of cholesterol metabolism is essential for maintaining liver function [31,32]. Moreover, proinflammatory cytokines have the potential to modify liver function and decrease cholesterol efflux and transport, thereby modulating cholesterol metabolism [33,34]. The close association of *Bifidobacterium* and cholesterol metabolism and the effectiveness of *Bifidobacterium* on cholesterol metabolism have been presented in various studies [20,35]. The effect of the *Bifidobacterium* IPLA R1 strain has been shown, which can up-regulate the gene Hmgcr. And the *Bifidobacterium* IPLA R1 strain encodes a rate-limiting enzyme in hepatic cholesterol synthesis, thereby modulating cholesterol metabolism [20]. Furthermore, multiple studies have indicated that *Bifidobacterium* species can produce EPS to reduce circulating cholesterol levels to regulate cholesterol metabolism [21]. In the present research, it is suggested that the alleviation of liver dysfunction in overweight or obese individuals by BBr60 may be linked to the regulation of cholesterol metabolism. Moreover, BBr60, as a potential probiotic for obesity management, has been found to significantly regulate serum metabolism and alleviate liver dysfunction and glycolipid metabolism disorder in overweight or obese individuals. However, the specific regulatory mechanism, whether dependent on gut microbiota, requires further investigation. Furthermore, additional research is required to assess the impacts of prolonged probiotic interventions and interventions administered at various time points, including during fasting periods.

## 5. Conclusions

In this trial, BBr60 significantly modified the metabolic profiles of overweight or obese young adults, impacting 134 serum metabolites and key pathways such as cholesterol and ascorbate and aldarate metabolism. BBr60’s role in regulating cholesterol metabolism suggests it can enhance liver function and aid obesity management by improving BMI. These results position BBr60 as a promising candidate for clinical metabolic health interventions in overweight and obese individuals. Of course, more in-depth exploration of the mechanism of probiotics on metabolic disorders in obese patients needs further mouse experiments. And further studies are also needed to evaluate the effects of longer probiotic interventions and interventions at different time points, such as fasting intake.

## Figures and Tables

**Figure 1 foods-13-03655-f001:**
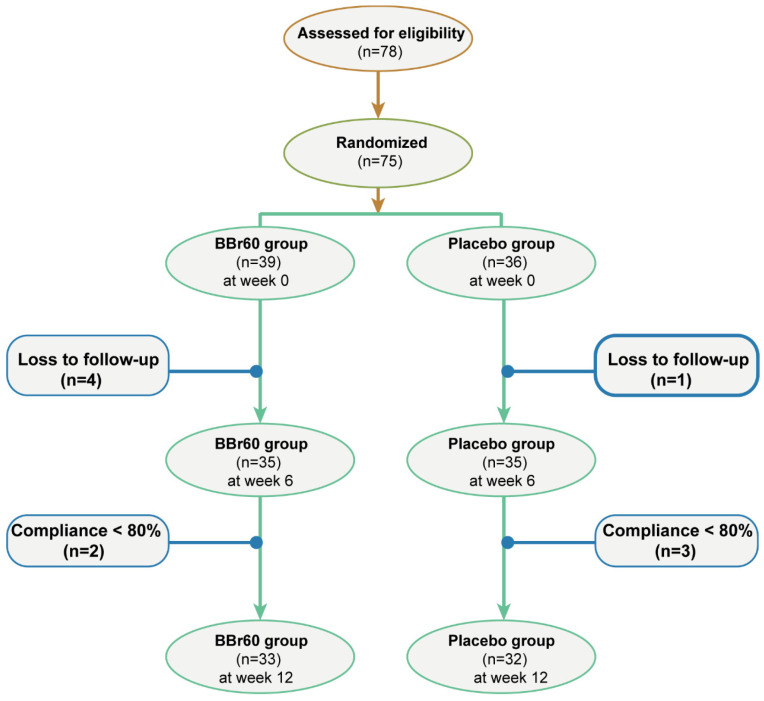
Selection flowchart for studies.

**Figure 2 foods-13-03655-f002:**
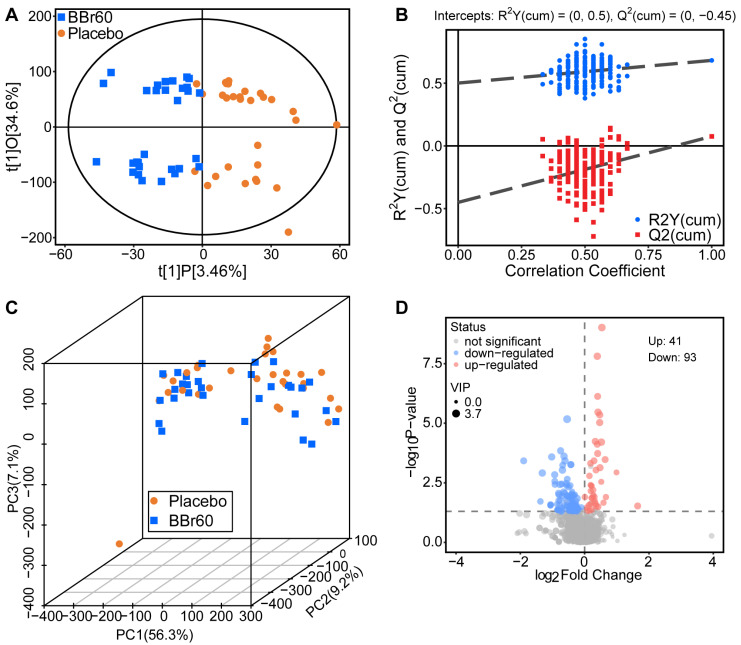
The effect of BBr60 on the serum metabolic profile in an adult population with overweight/obesity in the twelfth week. (**A**) A scores plot of OPLS-DA. (**B**) A permutation test of OPLS-DA. (**C**) A scores plot of PCA. (**D**) A volcano plot. Metabolites that are significantly up-regulated are shown in red, while those that are significantly down-regulated and non-significantly different are shown in blue or gray.

**Figure 3 foods-13-03655-f003:**
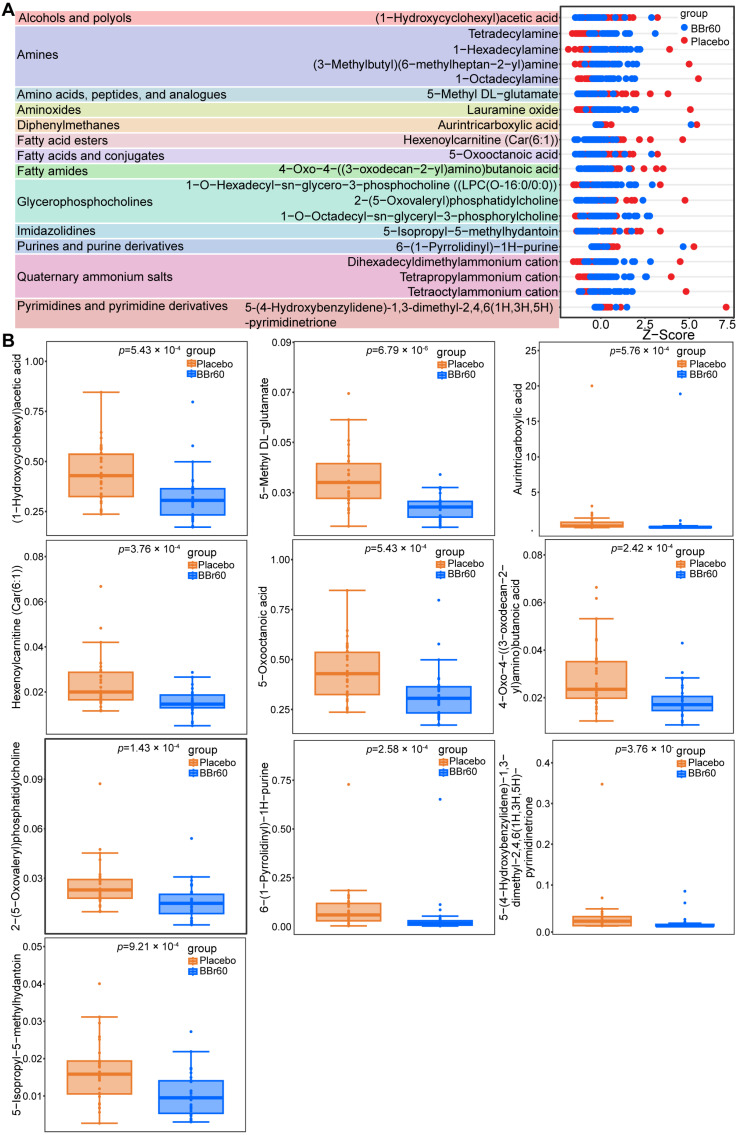
The effect of BBr60 on serum metabolites in the adult population with overweight/obesity in the twelfth week. (**A**) A Z-score plot of differential metabolites. (**B**) The relative abundance of significantly down-regulated metabolites. (**C**) The top ten most significantly up-regulated metabolites.

**Figure 4 foods-13-03655-f004:**
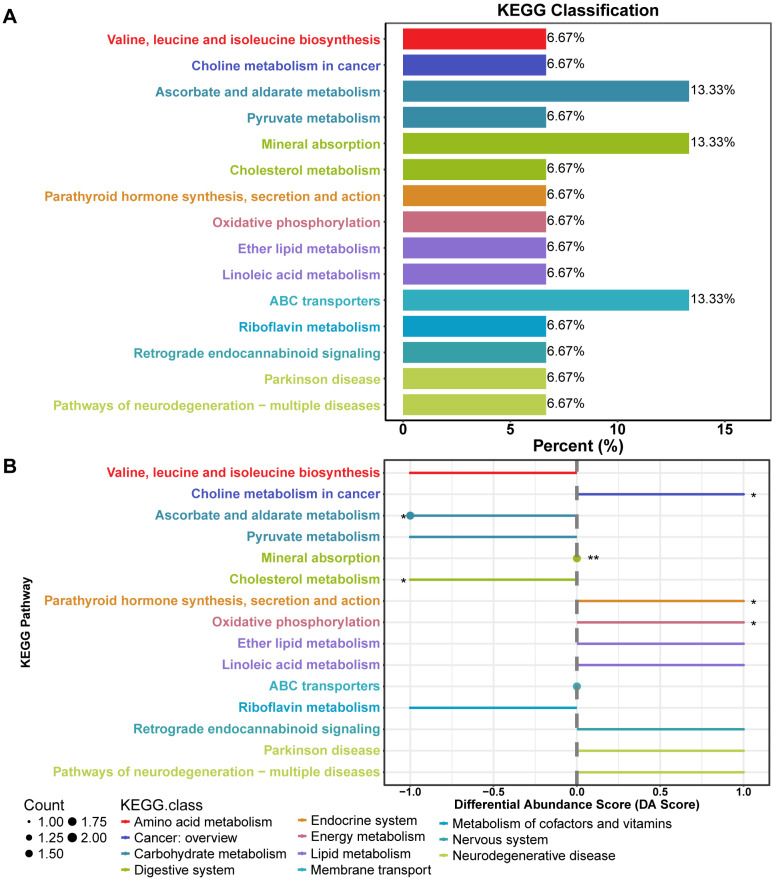
The effect of BBr60 on metabolic pathways in adult populations with overweight/obesity in the twelfth week. (**A**) A KEGG classification plot. (**B**) A differential abundance score plot, * *p*  <  0.05, ** *p*  <  0.01.

**Figure 5 foods-13-03655-f005:**
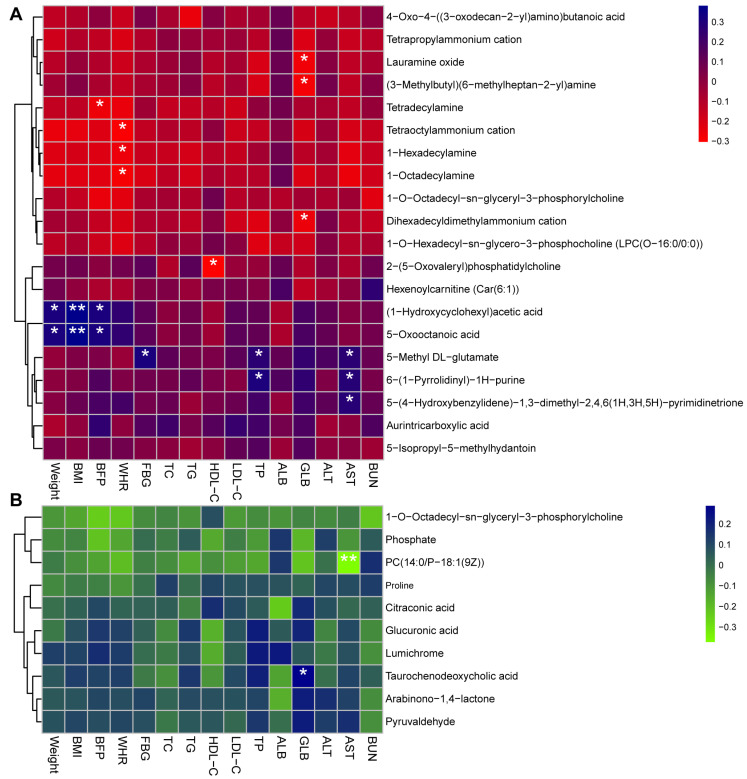
Correlation analysis between vital serum metabolites and clinic indictors in individual with overweight/obesity. (**A**) Correlation analysis between top 10 differentiating serum metabolites with significant up-regulation and down-regulation and clinic indexes in overweight/obesity. In R values, blue and red cells indicate positive and negative correlations, with * *p* < 0.05 and ** *p* < 0.01. (**B**) Analysis of correlation between serum metabolites linked to regulated metabolic pathways and clinical indices in individuals with overweight/obesity. Blue and green cells indicate positive and negative correlations, respectively; * *p* < 0.05 and ** *p* < 0.01. Abbreviations: body mass index (BMI); body fat percentage (BFP); Waist-to-Hip Ratio (WHR); fasting blood glucose (FBG); total cholesterol (TC); triglyceride (TG); high-density lipoprotein cholesterol (HDL-C); low-density lipoprotein cholesterol (LDL-C); alanine aminotransferase (ALT TG); aspartate aminotransferase (AST); total protein (TP); Albumin (ALB); globular proteins (GLB); blood urea nitrogen (BUN).

## Data Availability

The original contributions presented in the study are included in the article/Appendix A, further inquiries can be directed to the corresponding authors.

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
