# Peer review of "Modulation of Serum Metabolic Profiles by Bifidobacterium breve BBr60 in Obesity: A Randomized Controlled Trial"

_foods, 2024, doi:10.3390/foods13223655_

Round 1

Reviewer 1 Report

Comments and Suggestions for Authors

Overall, this paper presents a contribution to the field of obesity and probiotic research.

This is important to ensure that the metabolic changes observed are linked to significant phenotypic outcomes.

Major Point for Revision:

While the paper identifies metabolic pathways influenced by Bifidobacterium breve BBr60, the study lacks clarity regarding phenotypic differences between the BBr60 group and the control group, both before and after the intervention. 

Suggested Revision:

Figure 2: Principal Component Analysis (PCA) should present a comparison of all four groups:

  1. BBr60 group (before intervention)
  2. BBr60 group (after intervention)
  3. Control group (before intervention)
  4. Control group (after intervention)

This comparison would provide a clearer visual representation of whether there is a distinct separation between the BBr60 and control groups in terms of metabolic profiles. If no significant differences are observed in the PCA, it might suggest that the metabolomic changes do not translate into meaningful phenotypic variations. Adding these comparisons would enhance the understanding of the impact of BBr60 and substantiate the paper’s claims about its metabolic effects.

Author Response

Reviewer 1:

Overall, this paper presents a contribution to the field of obesity and probiotic research.

This is important to ensure that the metabolic changes observed are linked to significant phenotypic outcomes.

Major Point for Revision: While the paper identifies metabolic pathways influenced by Bifidobacterium breve BBr60, the study lacks clarity regarding phenotypic differences between the BBr60 group and the control group, both before and after the intervention.

Reply: Thank you for your valuable feedback and for highlighting the need for clearer delineation of phenotypic differences between Bifidobacterium breve BBr60 and the control groups in our manuscript titled “Modulation of Serum Metabolic Profiles by Bifidobacterium breve BBr60 in Obesity: A Randomized Controlled Trial.”. In response to your query, the focus of the current study was primarily on elucidating the metabolic changes induced by BBr60 in comparison to a placebo. We meticulously detailed these metabolic alterations and their implications on obesity-related clinical indicators. To further illustrate, we presented associations between specific serum metabolites (e.g., 5-Methyl DL-glutamate, 1-Hydroxycyclohexyl acetic acid) and clinical measures such as BMI and liver function markers, which reinforced BBr60’s potential role in modulating obesity-associated metabolic dysfunctions. Regarding the phenotypic changes, our data did show variations between the two groups post-intervention, which were manifested through changes in BMI, body fat percentage, and metabolic biomarkers like AST and total protein. These results signify BBr60's impact in improving the metabolic health of participants.

Furthermore, to address the complete scope of phenotypic and metabolic changes comprehensively, we have elaborated on these differences in another article[1], which study provides a detailed exploration of the gut microbiome and systemic metabolic modifications, thereby complementing and extending the findings reported in the current manuscript. We have amended our manuscript to include this cross-reference.

Suggested Revision: Figure 2 Principal Component Analysis (PCA) should present a comparison of all four groups:

  1. BBr60 group (before intervention)
  2. BBr60 group (after intervention)
  3. Control group (before intervention)
  4. Control group (after intervention)

This comparison would provide a clearer visual representation of whether there is a distinct separation between the BBr60 and control groups in terms of metabolic profiles. If no significant differences are observed in the PCA, it might suggest that the metabolomic changes do not translate into meaningful phenotypic variations. Adding these comparisons would enhance the understanding of the impact of BBr60 and substantiate the paper’s claims about its metabolic effects.

Reply: Thank you for your recommendation to incorporate a PCA that compares the metabolic profiles before and after the intervention for both the BBr60 and control groups. Following your suggestion, we have added supplementary Figure 1 to illustrate these comparisons across all four groups. Upon revising the figure and reanalyzing our data, we observed that the PCA did not demonstrate significant separations between the groups before and after the intervention. While at first glance this might seem to suggest minimal phenotypic variations, it is crucial to consider the complexities inherent in metabolomic studies. The lack of distinct clustering in PCA does not undermine the biological significance of the metabolic changes we observed. PCA serves as an exploratory tool that can be influenced by intragroup variability and may not always capture subtle yet biologically meaningful effects, particularly in multifactorial conditions such as obesity[2].

Importantly, despite the absence of clear separation in PCA, our study identified significant regulatory effects on specific metabolites and pathways, such as ascorbate and aldarate metabolism and cholesterol metabolism, which are known to play critical roles in obesity-related metabolic functions. This highlights that the metabolic modifications induced by BBr60, though not visually distinct in PCA, hold substantial biochemical significance. We have expanded the discussion in our manuscript to clarify the interpretation of PCA results in the context of metabolomic studies. Significant metabolic changes were found to be clinically relevant despite minimal PCA differences, supporting the notion that significant biological insights can be derived even in the absence of pronounced PCA clustering.

Reference

  1. Bai Z, Wu Y, Gao D, Dong Y, Pan Y, Gu S: Gut Microbiome and Metabolome Alterations in Overweight or Obese Adult Population after Weight-Loss Bifidobacterium breve BBr60 Intervention: A Randomized Controlled Trial. International Journal of Molecular Sciences 2024, 25(20):10871.
  2. Nyamundanda G, Brennan L, Gormley IC: Probabilistic principal component analysis for metabolomic data. BMC Bioinformatics 2010, 11(1):571.

Reviewer 2 Report

Comments and Suggestions for Authors

My comments are listed below:

The summary of results feels vague, instead of "regulating 134 serum metabolites and six metabolic pathways," specifying which pathways were affected and their relevance would make the abstract more informative.

The citation style is not in accordance with the journal guidelines

The introduction lacks a concise explanation of why Bifidobacterium breve was specifically chosen. More direct context would enhance readability and focus the reader's understanding of the study’s relevance.

In descriptions of statistical outcomes, terms like "obviously" and "effectively" are used subjectively. Use precise statistical language instead

e.g. "obviously down-regulated" seems subjective. How was the term “obvious” defined in this context, and does it reflect statistical significance or another threshold?

When discussing AST and serum metabolites, the statement "AST level showed obvious association with the content of PC(14:0/P−18:1(9Z)) associated with the pathways of linoleic acid metabolism, retrograde endocannabinoid signaling, choline metabolism in cancer" lacks specific correlation values. What were the exact correlation coefficients, and were these associations tested for significance individually or collectively?

The metabolic pathways of significance that were regulated included mineral absorption, ascorbate and aldarate metabolism, cholesterol metabolism, choline metabolism in cancer, parathyroid hormone synthesis, secretion and action, and oxidative phosphorylation," no prioritization is given to the pathways most relevant to obesity or metabolic health. Which of these pathways showed the strongest modulation, and how do they relate specifically to the context of obesity?

Some statements are overly generalized. For instance, "BBr60’s impact on serum metabolic profiles suggests it may improve liver function," could benefit from specifying how or why based on the study's findings.

Which specific liver function indicators improved in this study? Were changes in AST or ALT levels statistically or clinically significant? If liver function improvement is suggested, what mechanism might explain this outcome?

  The authors mentions cholesterol metabolism as one of the modulated pathways, implying potential health benefits. However, no specific link is drawn to clinical cholesterol levels. Did the study measure any traditional cholesterol markers (e.g., LDL, HDL, triglycerides) in addition to serum metabolites? If not, what practical conclusions can be drawn from the findings on cholesterol metabolism pathways alone?

The conclusion reiterates too much rather than synthesizing key takeaways. Instead of "The association analysis of the results revealed that BB60 may improve liver function in overweight or obese people by regulating cholesterol metabolism," it could be condensed to emphasize practical implications, like potential applications in clinical practice.

Author Response

Reviewer 2:

My comments are listed below:

  1. The summary of results feels vague, instead of "regulating 134 serum metabolites and six metabolic pathways," specifying which pathways were affected and their relevance would make the abstract more informative.

Reply: Thank you for your valuable feedback concerning the specificity of our results summary in the abstract. We agree that detailing the specific metabolic pathways affected by the BBr60 intervention and their relevance to obesity could greatly enhance the informativeness of the abstract. In response to your suggestions, we have revised the abstract to clearly specify the metabolic pathways that were significantly influenced by BBr60. We now explicitly mention that the intervention regulated crucial pathways such as ascorbate and aldarate metabolism, which plays a role in reducing oxidative stress; cholesterol metabolism, which is essential for lipid homeostasis; fatty acid biosynthesis, critical for energy storage; oxidative phosphorylation, for improved energy efficiency; and glycolysis/gluconeogenesis, pivotal in glucose metabolism. These enhancements aim to provide a clearer and more detailed overview of the metabolic impacts observed, highlighting their implications for managing metabolic health in obesity.

  1. The citation style is not in accordance with the journal guidelines

Reply: Thank you for pointing out the discrepancies in our citation style in relation to the journal's guidelines. We have carefully reviewed the citation format and have revised our manuscript to fully comply with the journal's prescribed style. All references have now been updated to ensure that they meet the formatting requirements specified in the journal's submission guidelines.

  1. The introduction lacks a concise explanation of why Bifidobacterium breve was specifically chosen. More direct context would enhance readability and focus the reader's understanding of the study’s relevance.

Reply: Thank you for your valuable feedback regarding the introduction of our manuscript. You noted the lack of a concise explanation for specifically choosing Bifidobacterium breve for our study. In response to your comment, we have revised the introduction to include a more direct and detailed explanation of why Bifidobacterium breve BBr60 was chosen. We clarified its documented efficacy in modulating gut microbiota, improving metabolic profiles, and reducing risk factors associated with obesity. We also highlighted previous studies demonstrating its impact on vital intestinal bacteria and its specific benefits in managing metabolic disorders, which are central to the objectives of our study.

  1. In descriptions of statistical outcomes, terms like "obviously" and "effectively" are used subjectively. Use precise statistical language instead e.g. "obviously down-regulated" seems subjective. How was the term “obvious” defined in this context, and does it reflect statistical significance or another threshold?

Reply: Thank you for your insightful feedback regarding the use of subjective terms like "obviously" and "effectively" in our manuscript. We understand the importance of precise statistical language in scientific communication. We have carefully reviewed the manuscript and revised all instances where subjective terminology was initially used to describe statistical outcomes. We have replaced these with specific statistical terms such as "statistically significant" and provided exact p-values where applicable to precisely reflect the data analysis results. These amendments ensure that the descriptions are now objectively based on the statistical evidence presented in our study.

  1. When discussing AST and serum metabolites, the statement "AST level showed obvious association with the content of PC (14:0/P−18:1(9Z)) associated with the pathways of linoleic acid metabolism, retrograde endocannabinoid signaling, choline metabolism in cancer" lacks specific correlation values. What were the exact correlation coefficients, and were these associations tested for significance individually or collectively?

Reply: Thank you for your valuable feedback requesting more precise statistical details in our discussion of the associations between AST levels and serum metabolites. We acknowledge the necessity of providing explicit correlation coefficients and significance testing to substantiate our findings. We have revised the relevant section to include exact correlation coefficients and p-values for each association mentioned. This addition not only clarifies the strength and significance of these relationships but also enhances the transparency and reproducibility of our results.

  1. The metabolic pathways of significance that were regulated included mineral absorption, ascorbate and aldarate metabolism, cholesterol metabolism, choline metabolism in cancer, parathyroid hormone synthesis, secretion and action, and oxidative phosphorylation," no prioritization is given to the pathways most relevant to obesity or metabolic health. Which of these pathways showed the strongest modulation, and how do they relate specifically to the context of obesity?

Reply: Thank you for your constructive feedback regarding the prioritization of the metabolic pathways in our study and their specific relevance to obesity. We have revised the discussion in our manuscript to more explicitly prioritize the pathways most relevant to obesity and metabolic health. We now highlight the pathways that showed the strongest modulation by BBr60 treatment and elucidate their specific roles in the context of obesity. We have emphasized that cholesterol metabolism, ascorbate and aldarate metabolism, and oxidative phosphorylation were the most significantly impacted pathways. These pathways are critical for managing obesity due to their roles in lipid homeostasis, oxidative stress reduction, and energy metabolism enhancement, respectively. This prioritization clarifies their relevance and underscores the potential of BBr60 in targeting metabolic disturbances associated with obesity.

  1. Some statements are overly generalized. For instance, "BBr60’s impact on serum metabolic profiles suggests it may improve liver function," could benefit from specifying how or why based on the study's findings.

Reply: Thank you for your insightful comment regarding the specificity of the statements in our manuscript, particularly the effects of BBr60 on liver function. We have revised the statement to provide a more precise explanation of how BBr60 impacts liver function based on our study's findings. We have clarified that the improvement in liver function was evidenced by the regulation of cholesterol metabolism, a pathway significantly modulated by BBr60, as well as by observed reductions in AST levels, which are indicative of enhanced liver health. These specific details from our results help substantiate the claim that BBr60 may improve liver function in obese individuals.

  1. Which specific liver function indicators improved in this study? Were changes in AST or ALT levels statistically or clinically significant? If liver function improvement is suggested, what mechanism might explain this outcome?

Reply: Thank you for your inquiry regarding the specific liver function indicators that showed improvement in our study, and the significance and mechanisms behind these changes. In our study, the liver function indicators that showed notable improvement were AST (aspartate aminotransferase) and TP (total protein). The reduction in AST levels was statistically significant, with p < 0.05, indicating a robust improvement in liver health post-intervention with BBr60. Although ALT (alanine aminotransferase) levels were also monitored, changes in ALT did not reach statistical significance. We have elaborated on these mechanisms in the discussion section of our manuscript, providing a detailed analysis of how BBr60's impact on these metabolic pathways could lead to improvements in liver function.

  1. The authors mention cholesterol metabolism as one of the modulated pathways, implying potential health benefits. However, no specific link is drawn to clinical cholesterol levels. Did the study measure any traditional cholesterol markers (e.g., LDL, HDL, triglycerides) in addition to serum metabolites? If not, what practical conclusions can be drawn from the findings on cholesterol metabolism pathways alone?

Reply: Thank you for your insightful question regarding the implications of modulated cholesterol metabolism pathways observed in our study, and the potential clinical correlations with traditional cholesterol markers such as LDL, HDL, and triglycerides. In this particular study, our primary objective was to investigate the broader serum metabolic profile changes following BBr60 intervention, focusing specifically on serum metabolites rather than traditional cholesterol markers. The rationale behind this approach was to explore underlying metabolic changes at a molecular level that could indicate potential pathways through which BBr60 exerts its effects, including but not limited to cholesterol metabolism. It is important to note that we have previously published findings related to traditional cholesterol markers (LDL, HDL, triglycerides) in a separate study[1], where we discussed the direct impact of BBr60 on these clinical cholesterol levels. The results of that study provided evidence of BBr60’s beneficial effects on lipid profiles, complementing the findings presented in this current manuscript. Therefore, while the current study did not measure traditional cholesterol markers directly, the modulation of cholesterol metabolism pathways observed herein supports the conclusions drawn in our previous research. This approach allows us to delineate the specific biochemical interactions influenced by probiotic treatment, offering a foundational understanding that supports practical conclusions about its potential health benefits. We have articulated this connection and rationale in the discussion section of our manuscript, aiming to provide a comprehensive overview of both the direct and indirect evidence of BBr60's impact on cholesterol metabolism and overall metabolic health.

  1. The conclusion reiterates too much rather than synthesizing key takeaways. Instead of "The association analysis of the results revealed that BB60 may improve liver function in overweight or obese people by regulating cholesterol metabolism," it could be condensed to emphasize practical implications, like potential applications in clinical practice.

Reply: Thank you for your constructive feedback on the conclusion of our manuscript. We have revised the conclusion to more concisely highlight the significant effects of BBr60 on the metabolic profiles of overweight or obese young adults, and its potential applications in clinical practice. The revised conclusion now succinctly states the regulatory impacts on crucial metabolic pathways and the corresponding clinical benefits, reinforcing BBr60's promise as a therapeutic agent.

Reference

  1. Bai Z, Wu Y, Gao D, Dong Y, Pan Y, Gu S: Gut Microbiome and Metabolome Alterations in Overweight or Obese Adult Population after Weight-Loss Bifidobacterium breve BBr60 Intervention: A Randomized Controlled Trial. International Journal of Molecular Sciences 2024, 25(20):10871.

Reviewer 3 Report

Comments and Suggestions for Authors

The manuscript ID foods-3284219 “Modulation of Serum Metabolic Profiles by Bifidobacterium breve BBr60 in Obesity: A Randomized Controlled Trial” has an “innovative purpose”. However, the present manuscript needs "major revision" before being accepted for "Foods (ISSN 2304-8158)".

Abstract: Authors should include numerical values ​​and/or percentages to better present the results. This assists the reader.

2. Materials and Methods:

Authors need to detail the sampling conditions (serum samples). What were the sample periods? (daily, only at the beginning and end of the experiment...).

What were the aseptic conditions for collection?

A qualified professional was responsible?

What is the volume of each sample?

Under what conditions were the samples stored?

How long were the samples stored? Were the samples analyzed on the day of collection or only after the end of the experiment?

Did all patients use the probiotic at the same time? What was the study recommendation?

All of this information is reflected in the discussion of the study. For this reason, this information also needs to be discussed.

5. Conclusions

The authors described the main results. However, they did not describe the future application of their studies.

Authors need to report the future prospects of the study. Is this study sufficient to indicate the use of the probiotic strain studied? Are new studies needed?

What is the daily recommendation for consumption of this probiotic to obtain results? How long can it be consumed? Is there a best time to consume? After meals or does the time not interfere?

Author Response

Reviewer 3:

The manuscript ID foods-3284219 “Modulation of Serum Metabolic Profiles by Bifidobacterium breve BBr60 in Obesity: A Randomized Controlled Trial” has an “innovative purpose”. However, the present manuscript needs "major revision" before being accepted for "Foods (ISSN 2304-8158)".

  1. Abstract: Authors should include numerical values ​​and/or percentages to better present the results. This assists the reader.

Reply: Thank you for your valuable feedback concerning the specificity of our results summary in the abstract. We added the numerical values in Line of 26, 31, 33.

  1. Materials and Methods:

Authors need to detail the sampling conditions (serum samples). What were the sample periods? (daily, only at the beginning and end of the experiment...).

 Reply: Thank you for your valuable feedback, this information has been supplemented in Line 127,128,129.

What were the aseptic conditions for collection?

Reply: Thank you for your valuable feedback, this information has been supplemented in Line 128.

A qualified professional was responsible?

 Reply: Thank you for your valuable question, a team consisting of one physician and several nurses of Hospital of Henan University of Science and Technology was in charge of blood extractions.

What is the volume of each sample?

 Reply: Thank you for your valuable question, the volume of each sample collected is 5 mL for serum metabolism analysis.

Under what conditions were the samples stored?

  Reply: Thank you for your valuable question, the serum will be transferred immediately to dry ice after collection, and then stored in a -80°C freezer after collection.

How long were the samples stored? Were the samples analyzed on the day of collection or only after the end of the experiment?

   Reply: Thank you for your valuable question, basic indictors (such as fasting blood glucose, total cholesterol) were examined immediately after collecting in Hospital of Henan University of Science and Technology. The serum from week 0 used for metabolism analysis was stored for 12 weeks until the end of the experiment at week 12. The serum of 0 and twelve weeks were sent together to the metabolism company for testing.

Did all patients use the probiotic at the same time? What was the study recommendation?

    Reply: Thank you for your valuable question, probiotics a day, it is recommended to take 30 minutes after meals, warm water or milk, water temperature does not exceed 40 degrees. Daily intake registration card needs to fill in the date and tick. The finished packaging should be retained and recycled. During the intake of probiotics, if you need to take antibiotics, need to interval 2-3 hours. The corresponding information is supplemented in Line of 116-121.

All of this information is reflected in the discussion of the study. For this reason, this information also needs to be discussed.

Reply: Thank you for your valuable feedback, we have refined the discussion section.

  1. Conclusions

The authors described the main results. However, they did not describe the future application of their studies.

 Reply: Thank you for your suggestion, we have described the future application of their studies in the part of Conclusion.

Authors need to report the future prospects of the study. Is this study sufficient to indicate the use of the probiotic strain studied? Are new studies needed?

 Reply: Thank you for your suggestion, we have added this part in Line of 377-381.

What is the daily recommendation for consumption of this probiotic to obtain results? How long can it be consumed? Is there a best time to consume? After meals or does the time not interfere?

Reply: According to the current study results, a daily intake of 1×1010 CFU daily for 12 weeks can be seen to have a significant effect. But whether there is a more pronounced effect if it lasts longer needs further study. In addition, regarding the timing of intake, the current study recommended that participants take it half an hour after a meal, but whether the effect is better when taking it on an empty stomach or before sleeping needs to be further evaluated.

Round 2

Reviewer 1 Report

Comments and Suggestions for Authors

My comments are solved in this revision. 

Reviewer 2 Report

Comments and Suggestions for Authors

The authors took into account the recommendations of the reviewers, I recommend the publication of the article

Reviewer 3 Report

Comments and Suggestions for Authors

The authors have performed all revisions. I consider the manuscript to be suitable for publication in its current form.